# Human–Computer Interaction with a Real-Time Speech Emotion Recognition with Ensembling Techniques 1D Convolution Neural Network and Attention

**DOI:** 10.3390/s23031386

**Published:** 2023-01-26

**Authors:** Waleed Alsabhan

**Affiliations:** College of Engineering, Al Faisal University, P.O. Box 50927, Riyadh 11533, Saudi Arabia; walsabhan@alfaisal.edu

**Keywords:** human–computer Interaction, 1D and 2D Convolution Neural Networks (CNN), speech emotion recognition (SER), EMO-DB, SAVEE, ANAD, BAVED datasets, LSTM, ensembling techniques

## Abstract

Emotions have a crucial function in the mental existence of humans. They are vital for identifying a person’s behaviour and mental condition. Speech Emotion Recognition (SER) is extracting a speaker’s emotional state from their speech signal. SER is a growing discipline in human–computer interaction, and it has recently attracted more significant interest. This is because there are not so many universal emotions; therefore, any intelligent system with enough computational capacity can educate itself to recognise them. However, the issue is that human speech is immensely diverse, making it difficult to create a single, standardised recipe for detecting hidden emotions. This work attempted to solve this research difficulty by combining a multilingual emotional dataset with building a more generalised and effective model for recognising human emotions. A two-step process was used to develop the model. The first stage involved the extraction of features, and the second stage involved the classification of the features that were extracted. ZCR, RMSE, and the renowned MFC coefficients were retrieved as features. Two proposed models, 1D CNN combined with LSTM and attention and a proprietary 2D CNN architecture, were used for classification. The outcomes demonstrated that the suggested 1D CNN with LSTM and attention performed better than the 2D CNN. For the EMO-DB, SAVEE, ANAD, and BAVED datasets, the model’s accuracy was 96.72%, 97.13%, 96.72%, and 88.39%, respectively. The model beat several earlier efforts on the same datasets, demonstrating the generality and efficacy of recognising multiple emotions from various languages.

## 1. Introduction

Emotion is a complex phenomenon that is influenced by numerous circumstances. Charles Darwin [1], one of the earliest scientists to study emotions, saw emotional expression as the last of behavioural patterns that had become obsolete owing to evolutionary change. In [2] theory of emotion was only a partial assertion and that feeling still serves an essential purpose; it is simply the nature of this purpose that has developed. Emotions are occasionally felt when something unexpected occurs, something to which evolution has not yet adapted, and in the circumstances such as these, emotional consequences start to take control. It is well-recognised that people’s emotional states can cause physical changes in their bodies. Emotions, for instance, may affect voice alterations [1]. Therefore, speech signals, which account for 38% of all emotional communication, can recognise and convey emotions [3]. The sound signals contain some elements that represent the speaker’s emotional state and those that correspond to the speaker and the speech. As a result, the fundamental idea behind emotion detection is examining the acoustic difference that arises while pronouncing the same thing in various emotional contexts [4]. Voice signals can be picked up without any device connected to the individual, even if alterations cannot be recognised without a portable medical instrument. Due to this, most studies on the subject have concentrated on automatically identifying emotions from auditory cues [5].

In human–computer interaction (HCI) and its applications, automatic emotion recognition is crucial since it can be a powerful feedback mechanism [6,7]. The main tools used in traditional HCI include the keyboard, mouse, screen, etc. It cannot comprehend and adjust to people’s emotions or moods; it just seeks convenience and precision. It is not easy to expect the computer to have the same intellect as people if it cannot recognise and convey emotions. Furthermore, it is challenging to anticipate that HCI will be genuinely harmonious and natural. People naturally expect computers to have emotional skills in the process of HCI since human interactions and communication are natural and emotional. The goal of affective computing is to give computers the capacity to perceive, comprehend, and produce a variety of emotional qualities similar to those found in humans. This will eventually allow computers to engage with humans in a way that is natural, friendly, and vivid [8].

Applications include human-robot communication, where machines react to people based on the emotions they have been programmed to recognise [9], implementation in call centres to identify the caller’s emotional state in emergencies, determining the degree of a customer’s satisfaction, medical analysis, and education. A conversational chatbot is another suggestion for an emotion recognition application, where real-time SER applications can improve dialogue [10]. A real-time SER should determine the best compromise between little computer power, quick processing times, and excellent accuracy.

But one of SER’s most significant yet elusive tasks has been identifying the “best” or most distinctive acoustic qualities that describe certain emotions. Despite thorough investigation, the pace of improvement was modest, and there were some discrepancies between studies. Due to these factors, research has shifted toward techniques that do away with or drastically minimise the requirement for prior knowledge of “optimal features” in favour of the autonomous feature-generating processes provided by neural networks [11]. Both passes employ cutting-edge classifiers, notably convolutional neural networks (CNN) and DNN [12,13,14].

Additionally, emotion identification may distinguish between feelings conveyed in only one language or across several languages. However, multilingual emotion recognition is still a new research topic, although many papers on monolingual emotion recognition have been published [15,16]. Therefore, using English, German, and Arabic corpora, extensive experiments and analyses of multilingual emotion recognition based on speech are given in the current study.

The contributions of the study are as follows:In this paper, two deep learning models for the SER are proposed: a 1D CNN with LSTM attention and a 2D CNN with multiple layers employing modified kernels and a pooling strategy to detect the sensitive cues-based input extracted features, which tend to be more discriminative and dependable in speech emotion recognition.The models above were constructed using varied datasets from multiple languages to make the models more generalisable across language boundaries in analysing emotions based on the vibrational patterns in speech.The performance of the proposed 1D CNN with the LSTM attention model is superior to that of the prior attempts on the various speech emotion datasets, suggesting that the proposed SER scheme will contribute to HCI.

The paper is organized as follows: the introduction is in Section 1, followed by the literature review in Section 2, then the dataset details, pre-processing, feature extraction, and methodology in Section 3, followed by the results in Section 4 and discussion in Section 5, and the conclusions is in Section 6, followed by the references.

## 2. Literature Review

Numerous studies have been done on extracting emotions from aural information. The SER system had two essential processes in standard Machine Learning techniques: manual feature extraction and emotion classification. This section discusses existing techniques that leverage various real-time speech emotion recognition datasets and feature extraction techniques available in the literature. Some of the existing models for real-time speech emotion recognition are also discussed in this section.

A flexible emotion recognition system built on the analysis of visual and aural inputs was proposed by [17] in 2017. Mel Frequency Cepstral Coefficients (MFCC) and Filter Bank Energies (FBEs) were two features used in his study’s feature extraction stage to lower the dimension of previously derived features. The trials used the SAVEE dataset and had a 65.28 percent accuracy rate.

In their suggested method [18], also used the spectral characteristics, or the 13 MFCC, to categorise the seven emotions with a 70% accuracy rate using the Logistic Model Tree (LMT) algorithm. Results from experiments using the Ryerson Audio-Visual Database of Emotional Speech and Song and the Berlin Database of Emotional Speech (EmoDB) (RAVDESS).

By integrating three models or experts, each focused on different feature extraction and classification strategy [19], achieved ensemble learning. The study used the IEMOCAP corpus, and a confidence calculation approach was suggested to overcome the IEMOCAP corpus’s data imbalance issue. The study found the key areas of crucial local characteristics work in conjunction with the attention mechanism to exploit each expert’s job from many aspects fully. Seventy-five percent accuracy was achieved [20].

The authors in [21] developed a new, lightweight SER model with a low computing overhead and good identification accuracy. The proposed methodology uses a straightforward rectangular filter with a modified pooling strategy that improves SER discriminative performance, as well as a Convolutional Neural Network (CNN) method to learn the deep frequency features. The proposed CNN model was trained using frequency features extracted from the voice data, and its ability to predict emotions was then assessed. The benchmark speech datasets for the proposed SER model were the interactive emotional dyadic motion capture (IEMOCAP) and the Berlin emotional speech database (EMO-DB). The evaluation results were 77.01 and 92.02 percent recognition, respectively.

A novel approach for SER based on the Bidirectional Long Short-Term Memory with Attention (BLSTMwA) model and Deep Convolutional Neural Network (DCNN) was developed by [8]. (DCNN-BLSTMwA). The speech samples are initially preprocessed using data improvement and dataset balancing. Second, as input for the DCNN, three-channel log Mel-spectrograms (static, delta, and delta-delta) are extracted. The segment-level features are then produced using the DCNN model that has already been trained on the ImageNet dataset. The unweighted average recall (UAR) for experiments using the EMO-DB and IEMOCAP databases was 87.86 and 68.50 percent, respectively [22,23,24].

In this work [25] used recurrent neural network (RNN) designs in their research. Their suggested model extracts relationships from 3D spectrograms across time steps and frequencies by combining parallel convolutional layers (PCN) with a squeeze-and-excitation network (SEnet), also known as PCNSE. Additionally, they used the Interactive Emotional Dyadic Motion Capture (IEMOCAP) and FAU-Aibo Emotion Corpus to demonstrate the viability of the suggested approach (FAU-AEC). On the IEMOCAP dataset, they successfully attained a weighted accuracy (WA) of 73.1 percent, an unweighted accuracy (UA) of 66.3 percent, and a UA of 41.1 percent on the FAU-AEC dataset [26,27,28].

Deep Convolutional Recurrent Neural Network was used by [29] to create an ensemble classifier for speech emotion recognition (SER). This paper introduces a novel method for SER tasks inspired by current work on speech emotion detection. They obtained 3-D log Mel-spectrograms of utterance-level log Mel-spectrograms along with their first and second derivatives (Static, Delta, and Delta-delta). They use deep convolutional neural networks to extract the deep features from 3-D log Mel-spectrograms and deep convolutional neural networks [26]. An utterance-level emotion is then produced by applying a bi-directional-gated recurrent unit network to express long-term temporal dependency among all features. An ensemble classifier employing Softmax and a Support Vector Machine classifier is used to increase the overall recognition rate [27]. On the RAVDESS (Eight emotional states) and Odia (Seven emotional states) datasets, the suggested framework is trained and tested. The experiment’s outcomes show that an ensemble classifier outperforms a single classifier in terms of performance. The degrees of accuracy attained are 77.54 percent and 85.31 percent [30,31].

A pre-trained audio-visual Transformer for understanding human behaviour was introduced by [32]. It was trained on more than 500k utterances from over 4000 celebrities in the VoxCeleb2 dataset. The model’s application in emotion recognition tries to capture and extract relevant information from the interactions between human facial and auditory activities. Two datasets, CREMAD-D (emotion classification) and MSP-IMPROV were used to test the model (continuous emotion regression). According to experimental findings, fine-tuning the pre-trained model increases the consistency correlation coefficients (CCC) in continuous emotion detection by 0.03–0.09 and the accuracy of emotion categorisation by 5–7 percent compared to the same model generated from scratch [33,34,35].

We recognize a pattern in the adoption of deep convolutional models that can teach from spectrogram descriptions of speech, per the literature study. There are numerous databases accessible, therefore choosing which databases to use for better training and validation is a difficult process. Although little research has been done on it, the attention mechanism can enhance the performance of SER systems. As a result, we suggested two models in this paper: the 1D CNN with LSTM attention model and the 2D CNN model. The results demonstrate that the proposed models outperformed the existing models in the literature. The brief description of the proposed model is present in the subsequent section.

## 3. Materials and Methods

This work aims to classify emotions based on the voice signal. The proposed methodology is illustrated in Figure 1. Based on the findings of individual studies. One cannot definitively state which classifier is superior for emotion recognition. The quality of the data has a direct bearing on the performance of the classifier. This is because the accuracy shifts depending on the characteristics of the data, such as the quantity, and density distribution of each class (emotions), and language as well [36]. In addition, the model can be trained using the features derived from the audio waves rather than feeding raw audio into a classifier. Since the features are more particular to the issue at hand, the model can be executed more effectively using the features that have been retrieved.

As shown in Figure 1, the first stage involved the extraction of features from the available labelled datasets, and the second stage involved the classification of the features that were extracted. ZCR, RMSE, and the renowned MFC Coefficients were retrieved as features. Then, two proposed models, 1D CNN combined with LSTM and attention and a proprietary 2D CNN architecture, were used for classification. After that, the evaluation of the performance of the proposed models is presented.

### 3.1. Datasets

This section briefly describes the datasets such as Arabic Natural Audio Dataset (ANAD) dataset, Basic Arabic Vocal Emotions Dataset (BAVED), The Surrey Audio-Visual Expressed Emotion (SAVEE) database, and The Berlin emotional speech database (EMO-DB).

#### 3.1.1. Arabic Natural Audio Dataset

The (ANAD) [37] dataset is comprised of three distinct feelings, namely, angry (A), surprised (S), and happy (H). The audio recordings that correlate to these feelings were collected from eight videos of live calls between an anchor and a human located outside the studio. Additionally, the audio files were taken from online discussion programs in Arabic. Eighteen manual labourers contributed to the creation of the ground truth for the videos that were collected. First, they assigned emotions such as anger, surprise, and happiness to each film. The final label is determined by taking the average of the ratings given by all eighteen of the labellers for each video. After that, each film is cut into sections titled “callers” and “receivers”. Next, the areas with laughter, noise, and silence were eliminated. After that, each chunk was automatically broken down into speech units that lasted one second. Figure 2 presents the results of the quantitative analysis performed on the corpus. Figure 2a demonstrates the distribution of emotions in the ANAD dataset. This figure clearly indicates the count of emotions in three distinct feelings, namely, angry (A), surprised (S), and happy (H).

#### 3.1.2. Basic Arabic Vocal Emotions Dataset 

Basic Arabic Vocal Emotions Dataset (BAVED) [38] is an audio/wave dataset including Arabic words spelled with varying levels of conveyed emotions. The dataset consisted of the recorded forms of seven Arabic words (0—like, 1—unlike, 2—this, 3—file, 4—good, 5—neutral, and 6—bad) across three emotional degrees. Level 0 is when the speaker expresses a low level of emotion, comparable to feeling exhausted or depressed. Figure 2b demonstrates the distribution of emotions in the BAVED dataset. Level 1 is the average level, where the speaker expresses neutral feelings; level 2 is when the speaker expresses strong positive or negative emotions (happy, joy, grief, wrath, etc.). The collection consists of 1935 recordings made by 61 speakers (45 males and 16 females).

#### 3.1.3. Surrey Audio-Visual Expressed Emotion Database

The Surrey Audio-Visual Expressed Emotion (SAVEE) database [39] has been recorded as a prerequisite for creating an automatic emotion identification system. Figure 2c demonstrates the distribution of emotions in the SAVEE dataset. The collection contains recordings of four male actors expressing seven distinct emotions: sadness, neutral, frustration, happiness, disgust, and anger. The dataset provided 480 British English utterances in total. The sentences were selected from the standard TIMIT corpus and balanced phonetically for each emotion. The data were recorded, analysed, and labelled using high-quality audio-visual equipment at a lab for visual media.

#### 3.1.4. Berlin Database of Emotional Speech 

The Berlin emotional speech database, also known as EMO-DB [9], is a public database that includes information on seven different emotional states. These states are anger, boredom, disgust, fear, happiness, neutral, and sadness. Verbal contents originate from 10 German (5 males and 5 females) pre-defined neutral utterances. It was decided to have ten professional actors read each utterance in each of the seven emotional states. Figure 2d demonstrates the distribution of emotions in the EMO-DB dataset which shows the count of emotions in the seven distinct types of feelings. The EMO-DB includes around 535 phrases representing each of the seven emotions. The sound files were captured at a sample rate of 16 kHz, a resolution of 16 bits, and a single channel. The length of time for each audio file is, on average, three seconds.

### 3.2. Feature Extraction

In the feature extraction process, the system extracts as much information as possible from each piece of audio data. For example, the current work extracts the three most relevant features ZCR, RMSE, and MFCC.

#### 3.2.1. Zero Crossing Rate (ZCR)

The zero-crossing rate indicates the frequency at which the signal crosses the zero-amplitude level. Equation (1) provides the calculation of ZCR
(1)$$\mathrm{ZCR}=\frac{1}{T}\sum_{t=1}^{T}\left|s\left(t\right)-s\left(t-1\right)\right|$$
where *s*(*t*) = 1 if the signal has a positive amplitude at *t*, otherwise 0.

#### 3.2.2. Root Mean Square Error (RMSE)

It is the measure of the energy content in the signal. It is the measure of the energy content in the signal (Equation (2))
(2)$$RMS_{t}=\sqrt{\frac{1}{K}\sum_{k=t.K}^{\left(t+1\right).\left(K-1\right)}s{\left(k\right)}^{2}}$$

#### 3.2.3. Mel Frequency Cepstral Coefficients 

Studies in the field of psychophysics have demonstrated that the way humans perceive the sound frequency contents of speech signals does not follow a linear scale. Therefore, a subjective pitch is assessed on a scale referred to as the ‘Mel’ scale. This scale is used for tones with an actual frequency, *f*, measured in hertz. The conversion of the actual frequency *f* to *mel* scale is given in Equation (3).
(3)$$mel=1127\;log\;\left(1+f/700\right)$$

Mel frequency cepstral coefficients, often known as MFCC, are the individual coefficients that, when added up, form a MFC. They are a form of the cepstral portrayal of the audio sample, where they originated (a nonlinear “spectrum-of-a spectrum”). The typical method for deriving MFCCs is illustrated in Figure 3. Following the pre-processing stage, the speech frame will go through a hamming window, and then, a rapid Fourier transformation will be used to determine the energy distribution. To get rid of the influence that harmonics have, a Mel filter bank is utilised. The discrete cosine transformation is the last step in the process. In a basic MFCC, the first difference parameters, the second difference parameters, and energy, respectively, make up an N-dimensional MFCC feature.

### 3.3. Model Development

The methods of classification that were employed are discussed in the following subsections. To improve SER, the 1D CNN with LSTM and attention layers were implemented. In addition to that, a 2D CNN is also demonstrated here. The models were trained using the ZCR, RMSE, and MFCC features retrieved from the data.

#### 3.3.1. One-Dimensional Convolution Neural Network with Long Short-Term Memory and Attention

As shown in Figure 4, the suggested model consists of five blocks of convolution operations and two completely connected layers. The input to the convolution was retrieved features from the unprocessed audio files that were scaled to fit into 1D convolutional layers. The amount of input varies depending on the utilised dataset. In addition, the weights of the convolution layer must be taught, whereas the pooling layers employ a fixed function to translate the activation. The Rectified Linear Unit (ReLU) is utilised to offer non-linearity to the entire network without affecting the receptive fields of the convolutional layer. Loops improve the effectiveness of the convolution layer. In addition, the model output is produced using a training dataset that includes the loss function, and the learning parameters (kernels and weights) are updated using backpropagation with the loss. Finally, the result is pooled, essentially a nonlinear spatial down-sampling procedure. Pooling reduces the representation’s spatial size, which helps decrease parameters and calculations, preventing overfitting.

In the second half, CNN’s output is sent into LSTM, determining the more profound temporal link between features. An additional attention layer was a way of acquiring additional information pertinent to each class. Lastly, the pooled feature map is remapped using two layers that are fully connected. Finally, the Softmax layer provided the likelihood of each class’s prediction.

#### 3.3.2. Two Dimensional Convolution Neural Network (CNN)

The structure of the 2D CNN was similar to the common CNNs for the image classification tasks, with the convolution and pooling layers alternatively appearing. However, the number of the convolution and the pooling layers were less compared to the 1D CNN. This is because the primary audio signals were 1D, and the extracted features were 1D, but in the 2D CNN presented in Figure 5, the convolution kernel, feature map, and other network structures are 2D. Therefore, to process a 1D signal with 2D CNNs, the 1D signal is usually mapped to a 2D space. Then, these 2D features are input into the conventional 2D CNNs for further processing. Finally, an additional dropout layer is added for better training and to avoid overfitting the model.

### 3.4. Performance Evaluation

The most common measures that are used to evaluate model performance are presented in this section. Accuracy alone will not be enough; additional metrics such as precision, recall, and the F1-score evaluation criteria will be necessary. The F1- measure is the harmonic mean of the precision, as described by definition [40].

## 4. Results

The experiments conducted to validate the effectiveness of our suggested 1D CNN with the LSTM attention model are shown in this section. To avoid making observations based solely on a corpus examination, four different well-known databases were selected to demonstrate the effectiveness of the suggested approach. An ablation study was first conducted to clarify the advantages of adding LSTM and an attention mechanism to the 1D CNN architecture. A comparison of the created framework’s effectiveness with the outcomes of the designed 2D CNN architecture on the well-known databases ANAD, BAVED, SAVEE, and EMO-DB further emphasises its efficiency.

From the audio samples of each database, the Zero Crossing Rate (ZCR), Root Mean Square Error (RMSE), and Mel-Frequency Cepstral Coefficients (MFCC) features were extracted. The ZCR, which is measured as the number of zero crossings in the temporal domain during one second, is one of the most affordable and basic characteristics. One of the methods most frequently used to assess the accuracy of forecasts is RMSE, also known as the root mean square deviation. It illustrates the Euclidean distance between measured true values and forecasts. The windowing of the signal, application of the DFT, calculation of the magnitude’s log, warping of the frequencies on a Mel scale, and application of the inverse DCT are the main steps in the MFCC feature extraction approach. 

Before feature extraction, the data were also enhanced with noise and pitch and provided for model training. The model per epoch evaluation of the model training reveals that the proposed models raise the degree of accuracy and decrease losses for the training and testing datasets, indicating the model’s importance and efficacy. However, some models were stopped early before reaching epoch 50 as the validation accuracy has not improved much. The suggested CNN model’s visual output is depicted in Figure 6, while the confusion matric acquired are displayed in Figure 7 followed and Figure 8 for Loss and accuracy plots. It displays the results of training the 1D CNN with LSTM attention and 2D CNN models for the four spoken emotion datasets. This indicates the training and validation accuracy and the loss for the BAVED, ANAD, SAVEE, and EMO-DB datasets. Figure 6 presents the loss and accuracy plots during the training process of the 1D CNN with LSTM attention for the different datasets of real-time speech emotion recognition.

The suggested model is analysed, and the analysis results are presented in Table 1, while the confusion matric acquired are displayed in Figure 9. The AUC-ROC curve, recall, precision, and accuracy of the proposed deep learning models can be measured with the use of a confusion matrix. Our model demonstrates accuracy, precision, and recall, as well as the F1-score, which reflects the model’s resilience straightforwardly and concisely. In addition, the performance of the 1D CNN model with LSTM attention layers is much better than that of the 2D CNN model.

Figure 8 presents the loss and accuracy plots during the training process of the 2D CNN for the different datasets of real-time speech emotion recognition.

The confusion matrix in Figure 7 and Figure 9 and Table 2, Table 3, Table 4 and Table 5 clearly explains the model performance in detecting individual emotions on the test dataset. 

The 1D CNN with the LSTM attention model achieved the highest level of performance when applied to the ANAD dataset. It was clear from Figure 7a and Table 2 that the best individual performance was achieved for the feeling “Angry” also using the 2D CNN model (Figure 9a; Table 2), but the performance of the model 1D CNN with LSTM was quite acceptable on the “Surprised” emotion. In contrast, the performance of the 2D CNN was much worse for the “Surprised” emotion state, which greatly damaged the model’s overall accuracy.

Compared to the ANAD dataset, the performance of the BAVED dataset was significantly lower, even though it was also an Arabic dataset. The two distinct models each had a distinct impact on the performance of their respective particular classes. The “Neutral” emotion class achieved the highest performance when the 1D CNN model was evaluated. However, the performance of the 2D CNN is superior for the “Low” emotion class (Table 3; Figure 7b and Figure 9b).

For the SAVEE datasets (Table 4; Figure 7c and Figure 9c), the performance of predicting “Happy”, “Surprised”, and “Neutral” emotions was highest with a 1D CNN model. In contrast, the performance of predicting these emotions with a 2D CNN model was very much lower. The performance was only improved for the emotion “Neutral” option, and even then, it was only 77% recall. Compared to the inferior emotion (“Fear”, with a recall of 95 percent) that could be predicted using 1D CNN, this prediction was much less accurate.

The trend seen in SAVEE can also be seen in the German dataset -EMO-DB. The performance of the 1D CNN with LSTM attention demonstrated a significant and noticeable difference. The emotions that performed the best with 1D CNN were “Disgust” and “Anger”. “Anger” still predominates on 2D CNN, but “Disgust” is relatively low. “Sad” performed better (Table 5; Figure 7d and Figure 9d).

## 5. Discussion

Speech Emotion Recognition is one of the contributing factors to human–computer interaction. The human brain processes the multimodalities to extract the spatial and temporal semantic information that is contextually meaningful to perceive and understand an individual’s emotional state. However, for machines, even though human speech contains a wealth of emotional content, it is challenging to predict the emotions or the speaker’s intention hidden in their voice [41]. 

### 5.1. Dataset Variability

For the SER challenge, several of the earlier efforts only included a single language dataset [42] and nevertheless managed to obtain excellent performance on that dataset. Despite its outstanding performance, the model was constructed using only one language, limiting its ability to comprehend various feelings. Due to this, the evaluation of the potential for generalisation, and consequently the real-world impact of the different techniques, is effectively constrained [43]. Robust embeddings for speech emotion identification should perform well across various languages to act as a paralinguistic cue [44]. However, human speech is so diverse and dynamic that no model can be reserved to be used forever [45].

Additionally, the diversity of languages causes an imbalance of available datasets for emotion recognition for minority languages compared to well-established majority languages such as English. This is particularly true for languages that are spoken by a smaller population. Despite this, it is necessary to develop a generalised model for multilingual emotional data for us to use the currently available datasets. The proposed model was validated by testing it on four datasets spanning three distinct languages, each with a different dataset size and a different category of emotions. Even though the performance was not very good on some datasets, the model tried to achieve a generalisation across different sets of emotions expressed in different languages.

### 5.2. Feature Extraction 

The current work was designed similarly to the traditional ML pipeline, with the manually extracted features contributing to the model classification. ZCR, RMSE, and the MFCCs were the features extracted. Although no Feature Selection has been performed, MFCCs have been selected because they are the most common representation in traditional audio signal processing, but the most recent works used spectrogram images as the input for developing deep learning models [46]. Some of them used raw audio waveforms also [47]. Raw waveforms and spectrograms avoid hand-designed features, which should allow better to exploit the improved modelling capability of deep learning models, learning representations optimised for a task. However, this incurs higher computational costs and data requirements, and benefits may be hard to realise in practice. Handcrafted features, even though incorporating huge manual tasks, understanding the features contributing to the classification task would be more efficient than manual feature extraction, especially for complex audio recognition tasks such as emotion recognition. The proposed work outperformed the work on image spectrograms, with an accuracy of 86.92% for EMO-DB and 75.00% for SAVEE datasets [46] and 95.10% for EMO-DB and 82.10% for SAVEE [48] (Table 6).

### 5.3. Classification

In this study, CNNs are used to construct the particular model with additional LSTM with attention. For example, the speech signal is a time-varying signal which needs particular processing to reflect time-fluctuating features [49]. Therefore, the LSTM layer is introduced to extract long-term contextual relationships. The results revealed that the 1D CNN model produced with LSTM had outperformed the model developed with 2D CNN architecture for all the datasets included in the study (Table 1). [50] followed the same pipeline as our study, where they retrieved manual features from the audio, such as MFCCs. They even included the feature selection method, but the Linear SVM classifier was used, but with a comparatively lightweight model, they obtained an accuracy of 96.02%, which is much closer to our findings. However, the proposed model was inferior to the model produced by [51] using the BAVED database. In another work [52] this technique is also applied. They used wav2vec2.0 and HuBERT as the feature extractors and the classifier head as the Multi-layer Perception Classifier coupled to a Bi-LSTM Layer with 50 hidden units. The performance is as shown in Table 7, Table 8 and Table 9.

The current effort sought to construct an SER capable of generalising across the language boundaries in speech emotion detection. The performance of the model was satisfactory. However, there is still an opportunity for further progress in some languages. In the future study, more extensive datasets from various disciplines will be focused on to get a better SER model for a more productive human–computer interaction. The field of human–computer interaction, especially voice-based interaction, is developing and changing virtually every day. The current work aims to construct artificially intelligent systems based on deep learning for recognising human emotions based on many modalities. In this paper, a unique 1D CNN architecture with LSTM and attention layers was created, as well as a 2D CNN, to recognise emotions expressed in Arabic, English, and German utterances. Before starting the model training process, the MFCC, ZCR, and RMSE features were retrieved. The learning models were implemented, and then, the outcomes of these models were exhibited using the audio dataset available in three languages. The proposed 1D CNN design with LSTM and attention layers recognised emotions with more precision than the 2D CNN architecture. The model’s accuracy was achieved at 96.72%, 97.13%, 96.72%, and 88.39% for the EMO-DB, SAVEE, ANAD, and BAVED datasets.

### 5.4. Limitations

The current work is performing well in terms of results and accuracy compared to other works, but it is observed that it is a little complex. The performance complexity evaluation is a bit heavy, and it can be resolved. Therefore, in the upcoming future, this can be taken into consideration for the smooth performance of the system.

## 6. Conclusions

The field of human–computer interaction, especially voice-based interaction, is developing and changing virtually every day. The current work aims to construct artificially intelligent systems based on deep learning for recognising human emotions based on many modalities. In this paper, a unique 1D CNN architecture with LSTM and attention layers was created, as well as a 2D CNN, to recognise emotions expressed in Arabic, English, and German utterances. Before starting the model training process, the MFCC, ZCR, and RMSE features were retrieved. The learning models were implemented, and then, the outcomes of these models were exhibited using the audio dataset available in three languages. The results revealed that the 1D CNN model produced with LSTM outperformed the model developed with 2D CNN architecture for all the datasets included in the study. The performance of the model was satisfactory; additionally, the performance of the 2D CNN was superior for the “Low” emotion class. The proposed model outperformed the work on image spectrograms, with an accuracy of 86.92% for EMO-DB, 75.00% for SAVEE [46], 95.10% for EMO-DB, and 82.10% for SAVEE. The proposed 1D CNN design with LSTM and attention layers recognised emotions with more precision than the 2D CNN architecture. The model’s accuracy was achieved at 96.72%, 97.13%, 96.72%, and 88.39% for the EMO-DB, SAVEE, ANAD, and BAVED datasets.

## Figures and Tables

**Figure 1 sensors-23-01386-f001:**
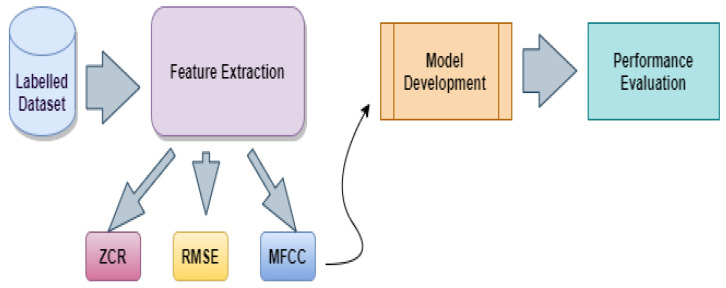
Proposed methodology.

**Figure 2 sensors-23-01386-f002:**
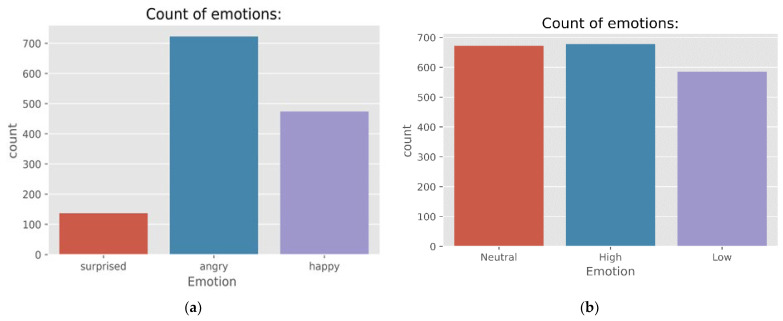
Distribution of emotions in the (**a**) ANAD, (**b**) BAVED, (**c**) SAVEE, and (**d**) EMO-DB datasets.

**Figure 3 sensors-23-01386-f003:**
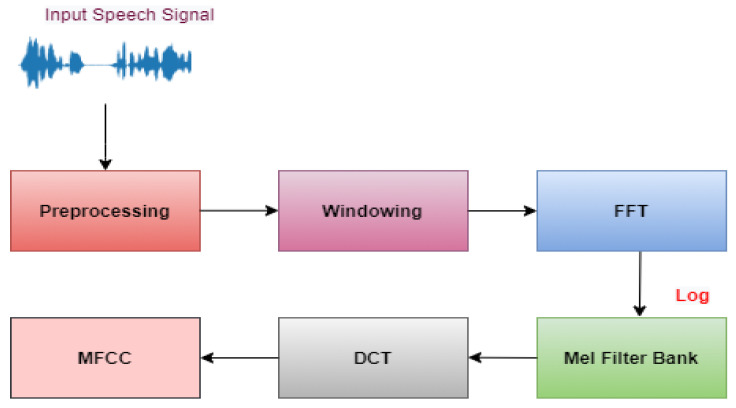
The MFCC production process.

**Figure 4 sensors-23-01386-f004:**
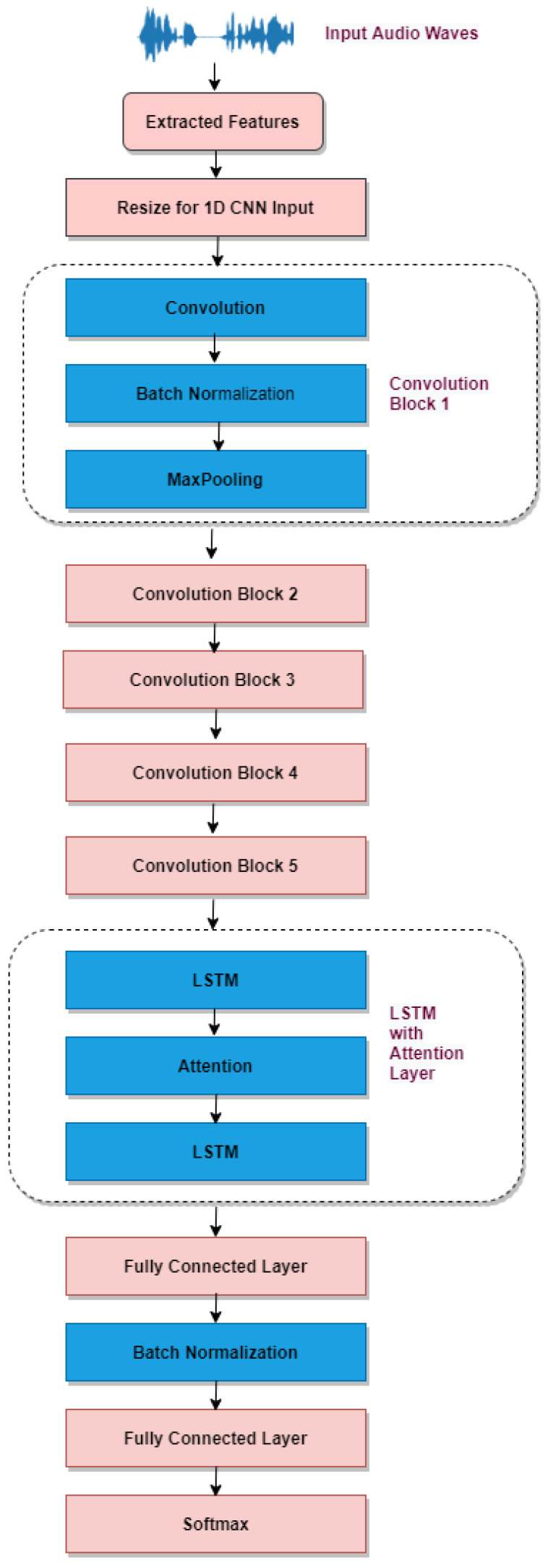
The 1D CNN with LSTM attention.

**Figure 5 sensors-23-01386-f005:**
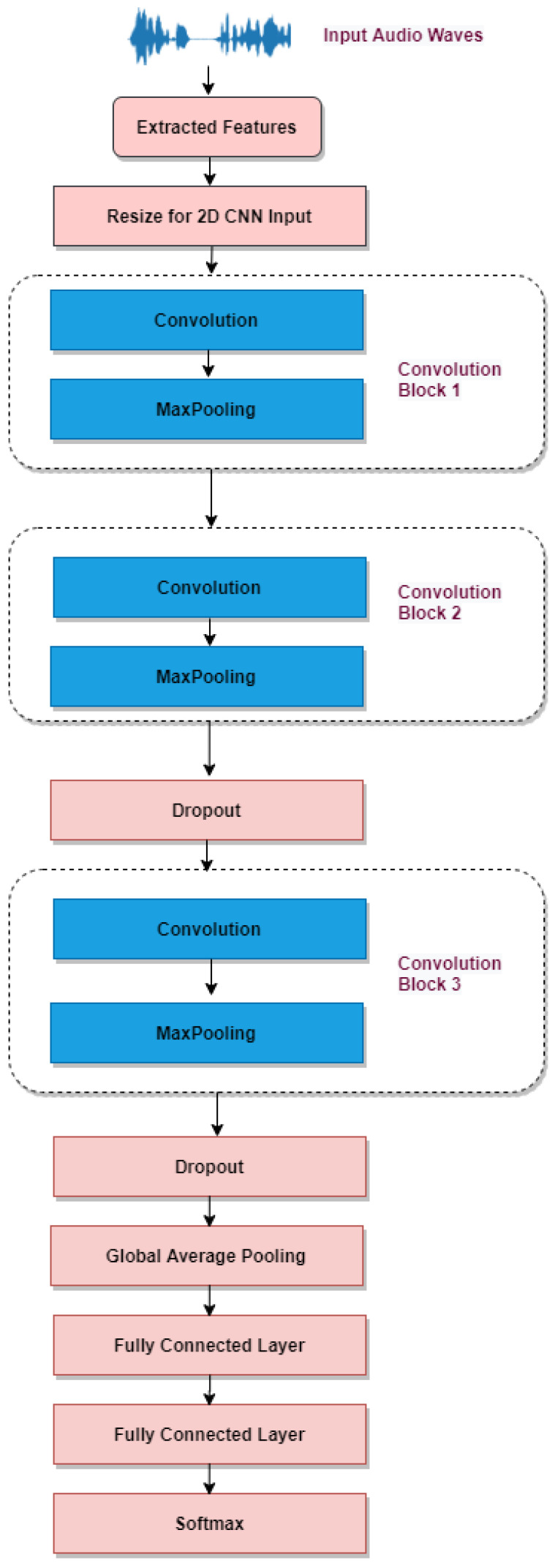
Two-Dimensional CNN.

**Figure 6 sensors-23-01386-f006:**
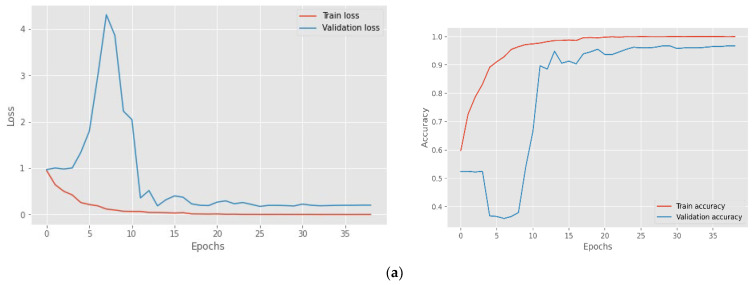
Loss and accuracy plots of the (**a**) ANAD, (**b**) BAVED, (**c**) SAVEE, (**d**) EMO-DB datasets for 1D CNN with the LSTM attention model.

**Figure 7 sensors-23-01386-f007:**
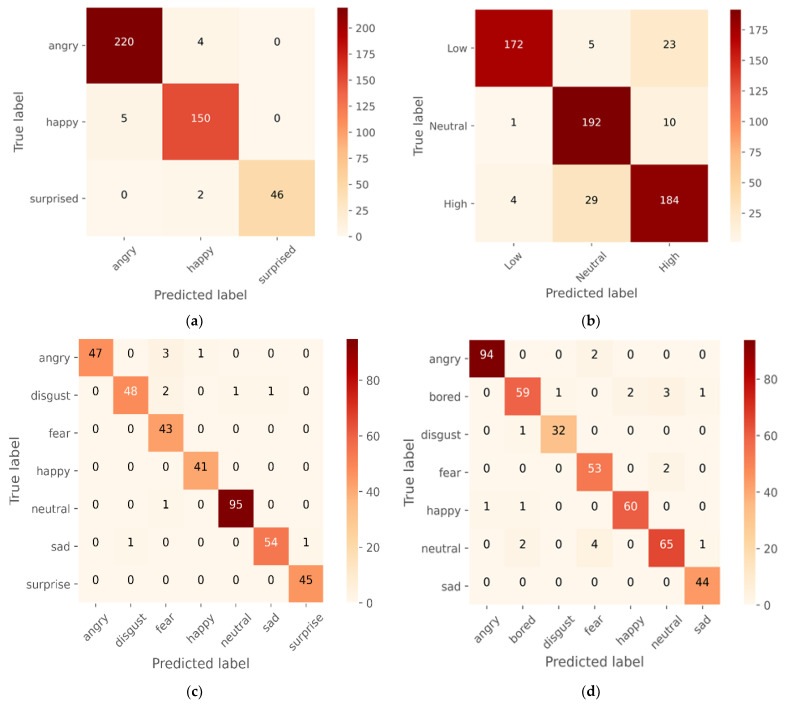
Confusion matrix of 1D CNN with the LSTM attention model for the (**a**) ANAD, (**b**) BAVED, (**c**) SAVEE, and (**d**) EMO-DB datasets.

**Figure 8 sensors-23-01386-f008:**
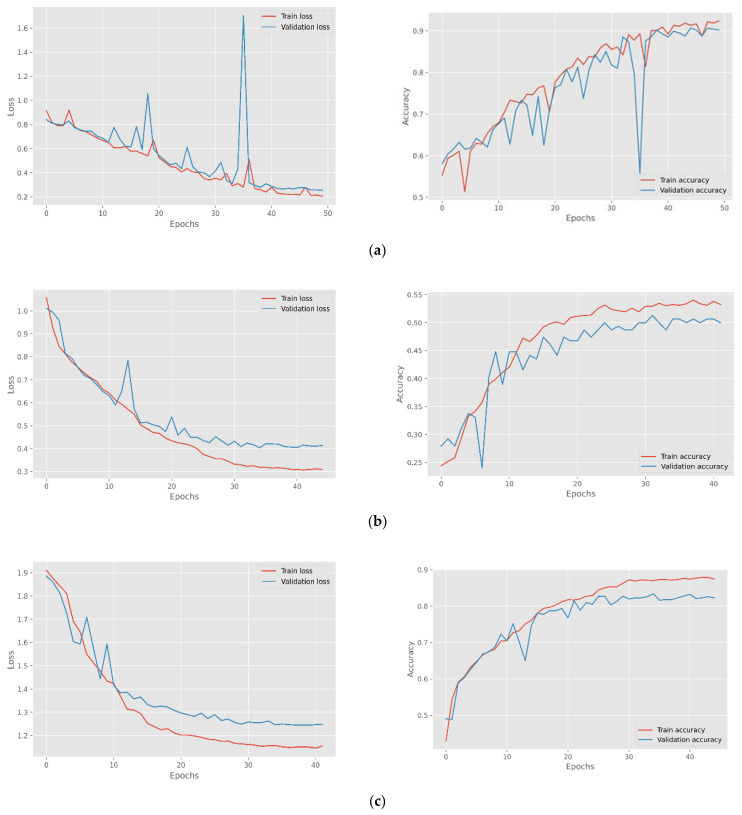
Loss and accuracy plots for the (**a**) ANAD, (**b**) BAVED, (**c**) SAVEE, and (**d**) EMO-DB datasets for the 2D CNN model.

**Figure 9 sensors-23-01386-f009:**
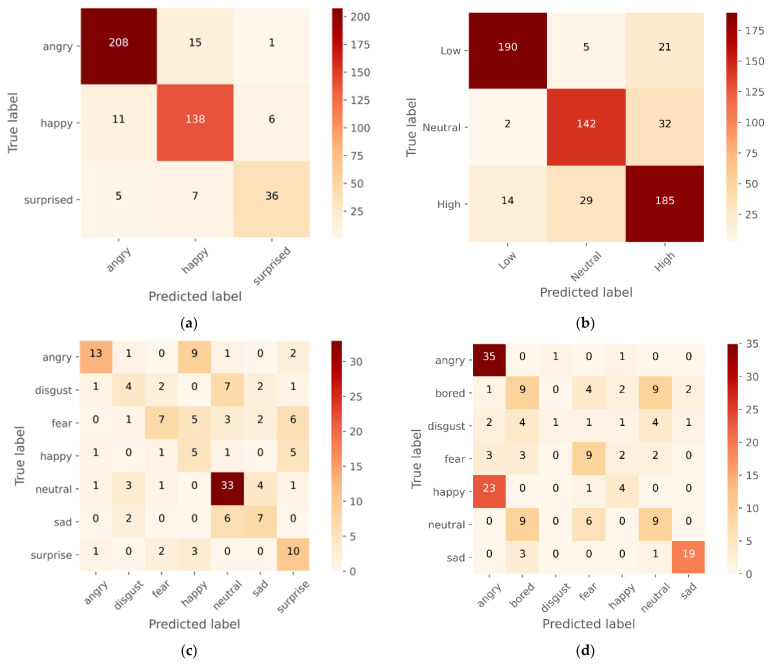
Confusion matrix of 2D CNN for the (**a**) ANAD, (**b**) BAVED, (**c**) SAVEE, and (**d**) EMO-DB datasets.

**Table 1 sensors-23-01386-t001:** Performance evaluation metrics.

Model	Dataset	Accuracy	Recall	Precision	F1-Score
2D CNN	ANAD	90.16%	85.79%	88.48%	86.94%
	BAVED	83.39%	83.26%	83.55%	83.38%
	SAVEE	51.30%	46.72%	48.68%	45.88%
	EMO-DB	50%	45.26%	48.86%	43.11%
CNN+LSTM+Attention	ANAD	96.72%	96.22%	95.64%	95.90%
	BAVED	88.39%	88.46%	88.97%	88.52%
	SAVEE	97.13%	97.12%	96.89%	96.91%
	EMO-DB	96.72%	96.96%	96.86%	96.88%

**Table 2 sensors-23-01386-t002:** Individual class performance evaluation metrics—ANAD.

Dataset	Model	Emotions	Recall	Precision	F1-Score
ANAD	CNN+LSTM+Attention	Angry	99%	97%	98%
		Happy	94%	98%	96%
		Surprised	96%	92%	94%
	2D CNN	Angry	93%	95%	94%
		Happy	92%	86%	88%
		Surprised	73%	85%	79%

**Table 3 sensors-23-01386-t003:** Individual class performance evaluation metrics—BAVED.

Dataset	Model	Emotions	Recall	Precision	F1-Score
BAVED	CNN+LSTM+Attention	Low	86%	97%	91%
		Neutral	95%	85%	90%
		High	85%	85%	85%
	2D CNN	Low	88%	92%	90%
		Neutral	81%	81%	81%
		High	81%	78%	79%

**Table 4 sensors-23-01386-t004:** Individual class performance evaluation metrics—SAVEE.

Dataset	Model	Emotions	Recall	Precision	F1-Score
SAVEE	CNN+LSTM+Attention	Angry	90%	98%	94%
		Disgust	98%	94%	96%
		Fear	95%	98%	96%
		Happy	100%	100%	100%
		Neutral	99%	97%	98%
		Sad	98%	96%	97%
		Surprise	100%	100%	100%
	2D CNN	Angry	50%	76%	60%
		Disgust	24%	36%	29%
		Fear	29%	54%	38%
		Happy	38%	23%	29%
		Neutral	77%	65%	70%
		Sad	47%	47%	47%
		Surprise	62%	40%	49%

**Table 5 sensors-23-01386-t005:** Individual class performance evaluation metrics—EMO-DB.

Dataset	Model	Emotions	Recall	Precision	F1-Score
EMO-DB	CNN+LSTM+Attention	Angry	100%	96%	98%
		Bored	95%	95%	95%
		Disgust	97%	97%	97%
		Fear	96%	95%	95%
		Happy	95%	100%	98%
		Neutral	97%	96%	97%
		Sad	93%	100%	96%
	2D CNN	Angry	95%	55%	69%
		Bored	33%	32%	33%
		Disgust	7%	50%	12%
		Fear	47%	43%	45%
		Happy	14%	40%	23%
		Neutral	38%	36%	37%
		Sad	83%	86%	84%

**Table 6 sensors-23-01386-t006:** Performance comparison of the works in the EMO-DB dataset.

Work	Model	Accuracy
[8]	DCNN-BLSTMwA	87.86%
[46]	VACNN+BOVW	86.92%
[48]	DCNN+CFS+SVM	95.10%
Our work	1D CNN with LSTM attention	96.72%

**Table 7 sensors-23-01386-t007:** Performance comparison of the works in the SAVEE dataset.

Work	Model	Accuracy
[46]	VACNN+BOVW	75
[48]	DCNN+CFS+SVM	82.1
Our work	1D CNN with LSTM attention	97.13%

**Table 8 sensors-23-01386-t008:** Performance comparison of the works in the ANAD dataset.

Work	Model	Accuracy
[50]	Linear SVM	96.02
Our work	1D CNN with LSTM attention	96.72%

**Table 9 sensors-23-01386-t009:** Performance comparison of the works in the BAVED dataset.

Work	Model	Accuracy
[51]	wav2vec2.0	89
Our work	1D CNN with LSTM attention	88.39%

## Data Availability

Data are available publicly and cited in proper places in the text.

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
