# Peer review of "Human–Computer Interaction with a Real-Time Speech Emotion Recognition with Ensembling Techniques 1D Convolution Neural Network and Attention"

_sensors, 2023, doi:10.3390/s23031386_

Round 1

Reviewer 1 Report

1. The overall structure and writing quality of the manuscript are poor.

2. The method proposed by the paper for multiple emotions recognition from various languages is not innovative. 

3. There are no detailed theories and algorithms described to support the experimental results.

Author Response

Response to Reviewers

Dear sir/madam

Thank you for giving us the opportunity to submit a revised draft of the manuscript “Human Computer Interaction with a Real-Time Speech Emotion Recognition with Ensembling Techniques 1D CNN and Attention” for publication in the Journal of MDPI. We appreciate the time and effort that you and the reviewers dedicated to providing feedback on our manuscript and are grateful for the insightful comments on and valuable improvements to our paper. We have incorporated most of the suggestions made by the reviewers. Those changes are highlighted within the manuscript. Please see below, in blue, for a point-by-point response to the reviewers’ comments and concerns. All page numbers refer to the revised manuscript file with tracked changes.

Reviewers' Comments to the Authors:

Reviewer #1

Comment 1: The overall structure and writing quality of the manuscript are poor.

Author Response:  We appreciate the reviewer’s feedback.  As suggested by the reviewer, we have revised the overall structure of the manuscript including figures, tables, sections, subsections, and references. Also, we have eliminated all the grammatical errors at our best in order to improve the English of the paper. Writing quality of the paper is now improved. Some major changes are:

The overlapping text and figures are now organized properly. Table’s columns and rows are properly adjusted.

Grammatical errors such as punctuations, Article uses, agreement mistakes, etc. are eliminated.

We have added description of section 2 (Literature Survey): “First, we discuss the datasets available in the literature related to real-time speech emotion recognition and then we discuss the feature extraction techniques available as well as used in the existing literature. Some of the existing models for real-time speech emotion recognition are also discussed in this section.”

Comment 2: The method proposed by the paper for multiple emotions recognition from various languages is not innovative.

Author Response:  We appreciate the reviewer’s feedback. We respectfully disagree. We proposed two deep learning models for the SER: a 1D CNN with LSTM attention and a 2D CNN with multiple layers employing modified kernels and a pooling strategy to detect the sensitive cues-based input extracted features, which tend to be more discriminative and dependable in speech emotion recognition. Both two deep learning models outperforms the available datasets. The proposed model outperformed the work on image spectrograms, with an accuracy of 86.92% for EMO-DB and 75.00% for SAVEE datasets [37] and 95.10% for EMO-DB, and 82.10% for SAVEE. The proposed 1D CNN design with LSTM and attention layers recognized emotions with more precision than the 2D CNN architecture. The model's accuracy was achieved at 96.72 %, 97.13 %, 96.72 %, and 88.39 % for EMO-DB, SAVEE, ANAD, and BAVED datasets.

Comment 3: There are no detailed theories and algorithms described to support the experimental results

Author Response:  We appreciate the reviewer’s feedback.  We added some detailed theories to back up the experimental study results. For the performance evaluation of the proposed models, we performed a performance comparison on various SER datasets. We went over these datasets in depth in both the literature review and the methodology section. We compared the proposed models to existing models in the literature to support the experimental results.

“The ZCR, which is measured as the amount of zero crossings in the temporal domain during one second, is one of the most affordable and basic characteristics. One of the methods most frequently used to assess the accuracy of forecasts is RMSE, also known as root mean square deviation. It illustrates the Euclidean distance between measured true values and forecasts. And, the windowing of the signal, application of the DFT, calcula-tion of the magnitude's log, warping of the frequencies on a Mel scale, and application of the inverse DCT are the main steps in the MFCC feature extraction approach.” (Added in result section 4)

Reviewer 2 Report

a). The contributions at the end of the introduction should be in bullets and not numbering

b). Figures need proper explanations

c). The methodology need detail descriptions to show what actually did and how the process was performed

d). Only diagrams have been taken as image but there is no proper descriptions that why were these drawn

e). Validity of the approach is missing, there should be proper description

f). Conclusions should show the details of the derivations drawn from the study.

Author Response

Human-Computer Interaction with a Real-Time Speech Emotion Recognition with Ensembling Techniques 1D CNN and Attention

Response to Reviewers

Dear sir/madam

Thank you for giving us the opportunity to submit a revised draft of the manuscript “Human Computer Interaction with a Real-Time Speech Emotion Recognition with Ensembling Techniques 1D CNN and Attention” for publication in the Journal of MDPI. We appreciate the time and effort that you and the reviewers dedicated to providing feedback on our manuscript and are grateful for the insightful comments on and valuable improvements to our paper. We have incorporated most of the suggestions made by the reviewers. Those changes are highlighted within the manuscript. Please see below, in blue, for a point-by-point response to the reviewers’ comments and concerns. All page numbers refer to the revised manuscript file with tracked changes.

Reviewers' Comments to the Authors:

Reviewer #2

Comment 1: The contributions at the end of the introduction should be in bullets and not numbering.

Author Response:  Thank you for pointing this out. The reviewer is correct, and we have added the bullets in place of numbering. The modification are highlighted in the revised manuscript.

Comment 2: Figures need proper explanations.

Author Response: As suggested by the reviewer, we have revised the explanations for all the figures. Modifications are highlighted in yellow in the revised manuscript. Some of the added explanations are:

Figure 1: “As shown in Figure 1, the first stage involved the extraction of features from the available labeled datasets, and the second stage involved the classification of the features that were extracted. ZCR, RMSE, and the renowned MFC Coefficients were retrieved as features. And then, two proposed models, 1D CNN combined with LSTM and attention, and a proprietary 2D CNN architecture was used for classification. After that, we evaluate the performance of the proposed models.” (Added in section 3)

Figure 2: “Fig.2 presents the results of the quantitative analysis performed on the corpus”.  (Added in section 3.1.1)

“Fig. 2a demonstrates the distribution of emotions in the ANAD dataset where you can clearly see the count of emotions in three distinct feelings, namely, angry (A), surprised (S), and happy (H).” (Added in section 3.1.1)

“Fig. 2b demonstrates the distribution of emotions in the BAVED dataset. Level 1 is the average level, where the speaker expresses neutral feelings; level 2 is when the speaker ex-presses strong positive or negative emotions (happy, joy, grief, wrath, etc.).” (Added in section 3.1.2)

“Fig. 2c demonstrates the distribution of emotions in the SAVEE dataset. The collection contains recordings of four male actors expressing seven distinct emotions: sadness, neutral, frustration, happiness, disgust, and anger.” (Added in section 3.1.3)

“Fig. 2d demonstrates the distribution of emotions in the EMO-DB dataset which shows the count of emotions in the seven distinct types of feelings.” (Added in section 3.1.4)

Comment 3: The methodology need detail descriptions to show what actually did and how the process was performed.

Author Response: We appreciate the reviewer’s feedback. We respectfully disagree. We went over each step of the proposed methodology in necessary detail. The datasets used in the manuscript for the study of real-time speech emotion recognition are discussed first, followed by the feature extraction technique used in the proposed model. Prior to feature extraction, the data was enhanced with noise and pitch and made available for model training. We extracted the Zero Crossing Rate (ZCR), Root Mean Square Error (RMSE), and Mel-Frequency Cepstral Coefficients (MFCC) features from each database's audio samples. The two proposed deep learning models for real-time speech emotion recognition are then discussed in detail.

Comment 4: Only diagrams have been taken as image but there are no proper descriptions that why were these drawn.

Author Response: We appreciate the reviewer’s feedback. The proper description for each diagram of the manuscript has been given. The diagram representing the process of proposed model as well as the performance evaluation of the proposed models.

Confusion matrix of Figures 7 and 9 are drawn for the evaluation of the AUC-ROC curve, recall, precision, and accuracy of our proposed deep learning models.

 “We analyzed the suggested model, and the analysis results are presented in Table 1, while the confusion matrices acquired are displayed in Figures 7 and 9. With the use of a confusion matrix, we can measure the AUC-ROC curve, recall, precision, and accuracy of our proposed deep learning models.” (Added in section 4- Results, page 11)

Fig. 6 presents the loss and accuracy plots during training process of the 1D CNN with LSTM attention for the different datasets of real-time speech emotion recognition. (added in section 4-Results, page-11)

 “Fig. 8 presents the loss and accuracy plots during training process of the 2D CNN for the different datasets of real-time speech emotion recognition.” (Added in section 4, page 12)

Comment 5: Validity of the approach is missing, there should be proper description.

Author Response: We appreciate the reviewer’s feedback. Our approach seems to be valid as we done the Performance comparison of the works with different dataset such as EMO-DB, SAVEE, ANAD, BAVED datasets in section 5. And our work outperformed these datasets. Table 6, 7, 8, and 9 (performance comparison with the existing works in the literature over different datasets) clearly demonstrate that our proposed work is valid and promising.

Comment 6: Conclusions should show the details of the derivations drawn from the study.

Author Response: As suggested by the reviewer, we have revised the conclusion section in order to show the details of the derivation drawn from the study. The added text:

“The results revealed that the 1D CNN model produced with LSTM had outperformed the model developed with 2D CNN architecture for all the datasets included in the study. The performance of the model was satisfactory also the performance of the 2D CNN is superi-or for the "Low" emotion class. The proposed model outperformed the work on image spectrograms, with an accuracy of 86.92% for EMO-DB and 75.00% for SAVEE datasets [37] and 95.10% for EMO-DB, and 82.10% for SAVEE.”

Reviewer 3 Report

The work is interesting and has promising results. However, the presentation style is needed to be revised, including the table format, misplaced text in the table, and some grammatical errors. Besides, the work still lacks a comparison regarding time complexity. Figure 1 is also too basic and simple. It is suggested to adjust the framework to be more specific and technological sound to the proposed model. 

Author Response

Human-Computer Interaction with a Real-Time Speech Emotion Recognition with Ensembling Techniques 1D CNN and Attention

Response to Reviewers

Dear sir/madam

Thank you for giving us the opportunity to submit a revised draft of the manuscript “Human Computer Interaction with a Real-Time Speech Emo-tion Recognition with Ensembling Techniques 1D CNN and Attention” for publication in the Journal of MDPI. We appreciate the time and effort that you and the reviewers dedicated to providing feedback on our manuscript and are grateful for the insightful comments on and valuable improvements to our paper. We have incorporated most of the suggestions made by the reviewers. Those changes are highlighted within the manuscript. Please see below, in blue, for a point-by-point response to the reviewers’ comments and concerns. All page numbers refer to the revised manuscript file with tracked changes.

Reviewers' Comments to the Authors:

Reviewer #3

Comment: The work is interesting and has promising results. However, the presentation style is needed to be revised, including the table format, misplaced text in the table, and some grammatical errors. Besides, the work still lacks a comparison regarding time complexity. Figure 1 is also too basic and simple. It is suggested to adjust the framework to be more specific and technological sound to the proposed model.

Author Response: Thank you for this suggestion. As suggested by the reviewer, we revised the paper's presentation style. We revised the table format and corrected some grammatical errors as well as misplaced text in the table. Figure 1 is overly basic and simplistic because it only depicts the proposed methodology's process. The detailed diagrams for the proposed model's elements are shown after the methodology section. We defined our proposed model in a step-by-step fashion in order to get the experimental results. Our proposed model is both specific and technologically sound.
